# Occurrence, risk factors and antimicrobial resistance of *Campylobacter* from poultry and humans in central Ethiopia: A one health approach

Fikre Zeru[1,2,3]*, Haileeyesus Adamu[1], Yohannes Hagos Woldearegay[2], Tesfaye Sisay Tessema[1], Ingrid Hansson[3], Sofia Boqvist[3]

**1** Institute of Biotechnology, Addis Ababa University, Addis Ababa, Ethiopia, **2** College of Veterinary Sciences, Mekelle University, Mekelle, Ethiopia, **3** Department of Animal Biosciences, Swedish University of Agricultural Sciences, Uppsala, Sweden

\* fikre.zeru@slu.se

## Abstract

### Background

*Campylobacter* is a leading foodborne pathogen posing a significant One Health challenge due to broad animal reservoirs and serious antibiotic resistance. Despite frequent human-animal-environment interactions in Ethiopia, One Health studies on *Campylobacter* occurrence and transmission are crucial but lacking.

### Methodology/Principal Findings

A cross-sectional study was conducted from March 2021 to March 2022 in central Ethiopia using a One Health approach to assess *Campylobacter* occurrence and resistance in humans, poultry and environment, and identify risk factors in humans and poultry. A total of 366 samples from 122 poultry farms were collected, including cloacal swabs, human stools, and poultry house floor samples. Epidemiological data on risk factors and respondents' awareness were gathered through interviews. *Campylobacter* spp. were isolated following ISO 10272, confirmed with multiplex PCR and tested for antimicrobial susceptibility using the disc diffusion method according EUCAST guidelines. *Campylobacter* spp. were detected in 12.5% samples, highest in poultry (19.6%), followed by humans (13.1%) and poultry house floor (4.9%). *Campylobacter jejuni* was the dominant species (80.4%), followed by *C. coli* (19.6%). In poultry, mixed farming with cattle increased *Campylobacter* colonization odds (adjusted odds ratio; AOR = 9.5), while all-in/all-out management decreased it (AOR = 8.4). In humans, *Campylobacter* infection was linked to raw milk consumption (AOR = 5.5), poultry access to living areas (AOR = 6.3), not using personal protective equipment when working with poultry (AOR = 8.3) and not washing hands after

**Data availability statement:** All data is available at Swedish University of Agriculture Sciences (SLU) repository. Data availability link: https://snd.se/en/catalogue/dataset/2025-54 DOI: https://doi.org/10.5878/bstc-9153.

**Funding:** The study was supported by the Swedish International Development Cooperation Agency (SIDA) through the research and training grant "Application of Biotechnology for environmentally safe and sustainable food security and green development of Ethiopia", a collaborative research project collaboration between Institute of biotechnology Addis Ababa University and the Swedish University of Agricultural Sciences (AAU-SLU) program: https://sida.aau.edu.et/index.php/international-comparative-education-phd-program/. The funder had no role in study design, data collection and analysis, decision to publish, or preparation of the manuscript.

**Competing interests:** The authors have declared that no competing interests exist.

handling poultry and cleaning barn (AOR = 5.6). Farm workers had a knowledge gap in zoonotic risks, including *Campylobacter* and One Health. High antibiotic resistance was observed, especially to erythromycin (63.0%), ciprofloxacin (69.5%), tetracycline (89.1%), and oxytetracycline (73.9%), with 69.5% of isolates showing multidrug resistance.

## Conclusions/Significance

The study revealed widespread occurrence of resistant *Campylobacter* spp. in poultry, workers, and the environment, highlighting the need for One Health interventions: improved biosecurity, hygiene, education, and responsible antimicrobial use to safeguard animal and human health.

## Author summary

A study conducted in and around Debre Berhan, central Ethiopia, analyzed samples from poultry farms for *Campylobacter* spp. and their antimicrobial resistance profiles. The highest prevalence of *Campylobacter* was found in poultry cloacal swabs, followed by human stool and poultry house floor samples. Poor farm biosecurity and management practices were linked to the occurrence of *Campylobacter* in poultry, while human infections were associated with raw milk consumption and inadequate hygiene practices by farm workers on poultry farms. The study emphasizes the zoonotic risks of *Campylobacter* and the need for a One Health approach to address its spread. *Campylobacter* isolates showed high resistance to commonly used antimicrobials, with many classified as multidrug-resistant. Co-occurrence of *C. jejuni* in poultry, farm workers, and the environment, all with similar multidrug-resistant patterns, suggests possible transmission among them. These findings highlight the widespread antimicrobial resistance in *Campylobacter* from poultry farms and the urgent need for responsible antibiotic use to limit the emergence of resistant strains. Given the serious implications of antimicrobial resistance, the zoonotic importance of *Campylobacter*, and the frequent human-animal-environment interactions in Ethiopia, it is crucial to implement a national plan for surveillance, prevention, and control, along with promoting rational antimicrobial use through a One Health approach.

## Introduction

*Campylobacter* is the leading foodborne pathogen associated with animal-source food products globally and an important zoonotic pathogen [1]. The rising global incidence of *Campylobacter* infections underscores the urgent need for greater attention to this growing health burden [2]. *Campylobacter jejuni* and *Campylobacter coli* have extensive animal reservoirs, commonly colonizing the intestines of food-producing and companion animals [3]. Poultry serves as a major reservoir for *Campylobacter*

and is a significant source of transmission to humans [4]. Infections also occur through direct contact with animal feces, environmental exposure, and food products other than poultry [4,5]. The wide distribution, multiple transmission routes, and serious antibiotic resistance [4] make *Campylobacter* a significant One Health concern.

*Campylobacter* is widespread in poultry farm environments, making farm management and biosecurity crucial for preventing transmission. Several risk factors for *Campylobacter* in broiler flocks have been identified in Sweden and Denmark, including proximity to other livestock, poor general tidiness at the farm, insufficient hygiene and biosecurity practices, old broiler houses, and high broiler age at slaughter [6,7].

As stated by Van Boeckel et al. [8], globally, the proportion of antimicrobials showing resistance above 50% increased from 0.15 to 0.41 in chickens. Resistant *Campylobacter* strains have been recognized as a serious public health threat by global stakeholders such as the WHO and CDC [9]. Due to its zoonotic nature, *Campylobacter* is exposed to antibiotics used in both human healthcare and veterinary practice, presenting a clear and imminent challenge to the One Health approach. The bacterium rapidly evolves under antibiotic pressure, adapting to various hosts and environments, which leads to multidrug-resistant variants. Additionally, *Campylobacter* can acquire antimicrobial resistance (AMR) determinants from other bacteria through horizontal gene transfer [10]. *Campylobacter* typically does not cause clinical signs in animals. Antibiotics are often used prophylactically in animal production, and this irresponsible and unregulated use of antibiotics has been found to be a major driver of AMR and contributing to the development of resistant strains and complicating the treatment of human infections [11]. Global data on *Campylobacter* shows that in low- and middle-income countries, the highest resistance rates are found for tetracycline (60%) and quinolones (60%) [8]. According to a review by Zenebe et al. [12], which compiled findings from various studies conducted in Ethiopia, antimicrobial resistance rates in *Campylobacter* ranged from 0% to 100%. The review reported that human isolates exhibited particularly high resistance to cephalothin (83%), amoxicillin (80%), and tetracycline (56.8%), along with moderate resistance to amoxicillin-clavulanate (36%), trimethoprim-sulfamethoxazole (32%), clindamycin (31%), and ceftriaxone (28%). Among animal isolates, the highest resistance, as noted in the same review, was observed to penicillin (100%), ciprofloxacin (71.2%), tetracycline (69.2%), cephalothin (60%), erythromycin (55%), and trimethoprim-sulfamethoxazole (39%).

Poultry farming is a vital sector in Ethiopia's agriculture, with approximately 57 million chickens, predominantly indigenous breeds in rural area and improved commercial breeds in urban and peri-urban areas of central Ethiopia [13]. However, challenges such as inadequate disease prevention and control, limited access to quality feed, and low technical knowledge among producers affect productivity [14]. Few studies have been published regarding *Campylobacter* prevalence in humans, animals, and food in Ethiopia [12]. Despite its critical role as a zoonotic pathogen, understanding the prevalence and transmission dynamics of *Campylobacter* in poultry farms in Ethiopia is limited. With frequent interactions among humans, animals, and the environment, comprehensive One Health studies are also essential but still insufficient. This study aimed to assess the occurrence of *Campylobacter* and its antimicrobial resistance profiles in humans, poultry, and the poultry environment in central Ethiopia, and to evaluate associated risk factors in humans and poultry using a One Health approach.

## Materials and methods

### Ethical considerations and approval

A formal letter of support was subsequently obtained from Addis Ababa University Institute of Biotechnology (AAU-IOB) to communicate with the Agricultural Offices in each of the selected districts. The purpose of the study was explained to all participants, emphasizing that their participation was voluntary, and that all data would be handled anonymously although the results derived from this study would be published. Informed verbal consent was obtained from all participants before the commencement of data collection. The study received ethical approval from the Institute of Biotechnology Institutional Review Board (Ethical Approval no. IOB/LB/2016/2024).

## Study area and setting

The study was conducted in and around Debre Berhan town in central Ethiopia (Fig 1), approximately 120 km northeast of Addis Ababa. Debre Berhan serves as the capital, administrative and economic center of the North Shewa Zone in the Amhara region. The area has a typical subtropical highland climate, characterized by two distinct seasons: the wet season, from June to September, and the dry season, from October to May. The average annual humidity in the area is recorded at 62.7% [15]. The local farming system is mixed system, involving both crop cultivation and livestock rearing. Poultry production plays a significant role in the region, serving as an important source of meat and eggs for the local population. Like other parts of central Ethiopia [14], commercial poultry farming is expanding in the urban and peri-urban areas of Debre Berhan.

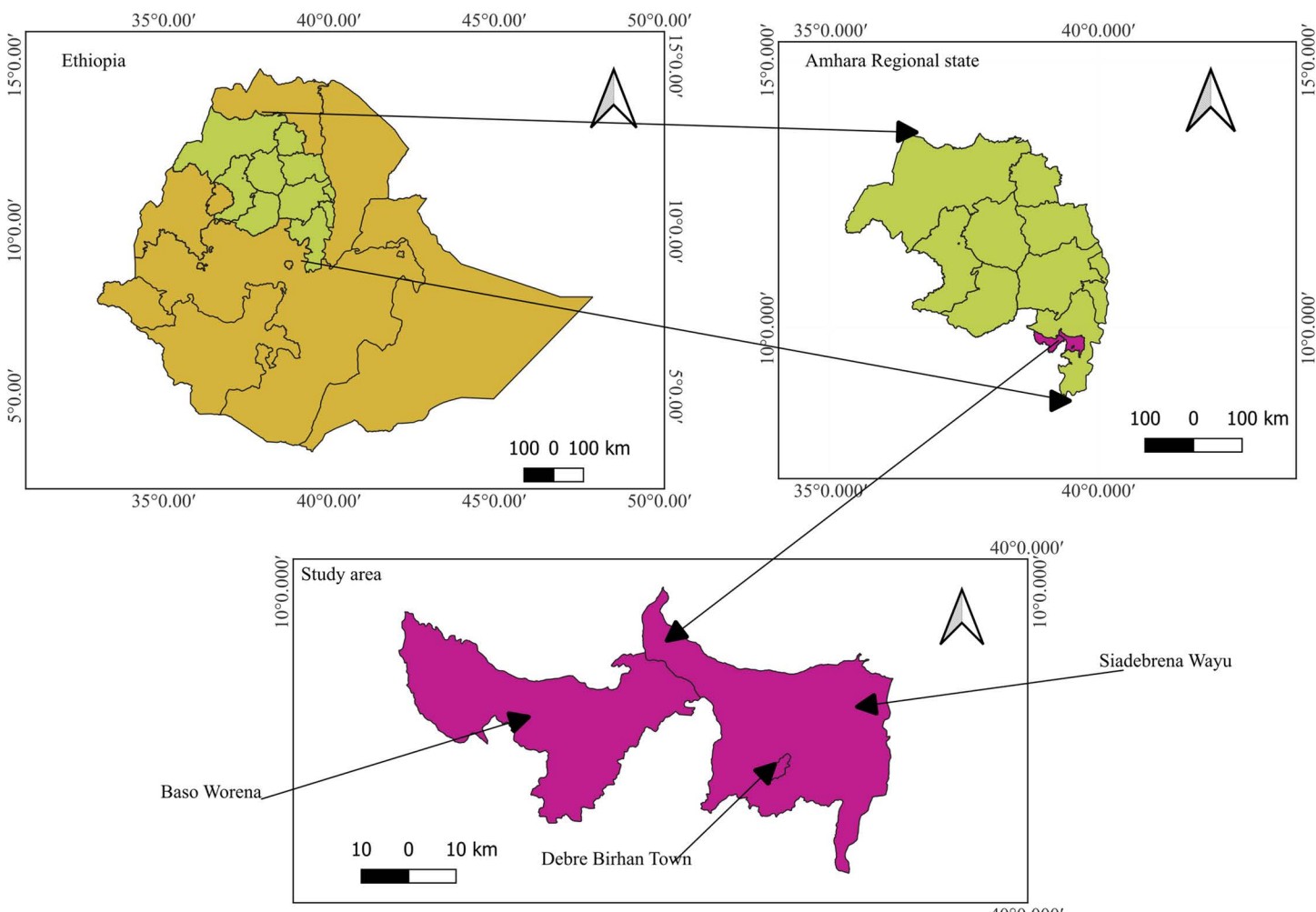

**Fig 1. Map of the study area: Debre Berhan town, Siadebrina Wayu, and Baso Worena Woredas.** Map created by the authors using QGIS v3.34.12. Base shapefiles were obtained from DIVA-GIS (https://www.diva-gis.org), derived from the GADM database. The data is publicly available for academic use and intended for non-commercial purposes.

## Study approach

A cross-sectional study was conducted from March 2021 to March 2022 on selected poultry farms, employing a One Health approach that integrated humans, animals, and the environment. The study districts were chosen due to their high concentration of poultry farms, including Debre Berhan town, Baso Worena, and Siadebrina Wayu. The study included urban (Debre Berhan town), peri-urban (outskirts of Debre Berhan town and Siadebrina Wayu), and rural (Baso Worena) areas. Before the study commenced, researchers visited the Agricultural Offices in each selected district to obtain approval and compile a list of poultry farms for the sampling frame. A veterinary professional in each district, who was familiar with the locations, was enlisted to assist in identifying poultry farms for inclusion in the study. The sample size was calculated using EpiInfo version 7.2.5.0 (Centers for Disease Control and Prevention, USA), with an expected prevalence of 50%, a 95% confidence level, a 5% absolute precision, and a design effect of 1.0. The finite population correction was applied based on the 159 poultry farms listed by the Agricultural offices. Initially, 113 poultry farms were planned to be included in the study; however, the sample size was increased to 122 to enhance the statistical power and reliability of the results obtained. The inclusion criteria for poultry farms were being either smallholder or commercial farms located in urban, peri-urban or rural areas of the study region, along with accessibility and willingness of the farmers to participate in the study. One sample was collected from each One Health domain at every farm, resulting in three samples per farm. Specifically, one chicken sample was collected using the grab sampling method, in which a chicken was randomly caught without prior selection based on size, color, or location. One human sample was also obtained per farm. If more than one human was present, a lottery method was used: everyone was assigned a unique number, and one was selected through a random draw. Lastly, one environmental sample was collected from the surface of the poultry house floor (Fig 2).

## Data collection

**Epidemiological data collection.** Epidemiological data were collected from the same individual who provided the human sample, using a structured questionnaire. The questionnaire was designed to assess potential risk factors associated with *Campylobacter* infection and its transmission, as well as the respondent´s general awareness of zoonotic diseases. The questions included a mix of open, closed, and semi-closed formats. The questionnaire was organized into four sections that addressed: (1) farm practices associated with *Campylobacter* risk factors in poultry; (2) consumption habits, occupation, and hygiene practices related to human *Campylobacter* infection; (3) poultry house conditions, and (4) awareness of poultry farm workers regarding *Campylobacter* as a Zoonosis and the One Health concept. The questionnaire was administered by the same interviewer through face-to-face interviews, with oral translation into the local language (Amharic) during the interviews to facilitate better understanding. The questionnaire was pretested on ten poultry farms located in the study area, selected based on accessibility and willingness to participate. These farms were

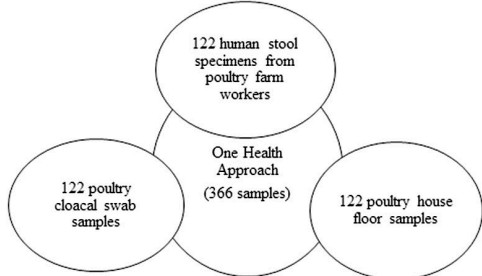

**Fig 2. One Health sampling approach and distribution of 366 samples collected from poultry, humans, and the environment on selected poultry farms around Debre Berhan, Ethiopia, for *Campylobacter* investigation.**

not included in the final study sample. Feedback from the pre-test was used to improve the clarity, relevance, and flow of the questions.

**Sample collection and transportation.** Approximately 10 grams of self-collected human stool specimens were placed into sterile 50 ml screw-capped jars containing Cary-Blair Transport Medium (Oxoid, Basingstoke, UK). Individuals were given two hours to provide the sample if unable to provide it immediately. Cloacal swab samples were collected from randomly selected poultry on the farm by a veterinarian using sterile cotton swabs, which were gently inserted into the cloaca. Immediately after sampling, the cotton swabs were transferred into Amie's agar gel with charcoal (Copan Diagnostics, Inc., Murrieta, CA, USA) for transport to the laboratory. Sock samples were collected from the poultry house floor using boot sock sampling, as described by Hansson et al. [16]. Sterile boot socks, moistened with Cary-Blair transport medium, were worn over the footwear, and the sampler walked the longest distance from wall to wall at least four times to cover all areas of the poultry house floor. After sampling, each sock was placed into a clean plastic bag with 10 ml of Cary-Blair Transport Medium, then securely sealed and labeled for identification.

All collected samples were placed in an icebox with ice packs until fieldwork was completed (up to 6 hours). They were then stored in a refrigerator at +4°C at the veterinary clinic of the respective district for a maximum of 24 hours. Subsequently, the samples were shipped in an icebox with ice packs to the Health Biotechnology Laboratory of the IOB-AAU, which took 2–3 hours. Upon arrival at the laboratory, the samples were processed immediately.

## Isolation of *Campylobacter* species

The analyses of *Campylobacter* were carried out using a modified version of ISO 10272–1:2017 [17]. Human stool and cloacal swab samples, expected to have high concentrations of *Campylobacter*, were directly plated onto modified Charcoal Cefoperazone Deoxycholate Agar (mCCDA) (Oxoid, Basingstoke, UK) supplemented with CCDA Selective Supplement (SR0155E, Oxoid, Thermo Scientific Inc, Basingstoke, UK). For poultry house floor sock samples, which were expected to have lower concentrations of *Campylobacter* and higher levels of background microflora, enrichment was performed in 90 ml of buffer peptone water without selective antibiotic supplement (HiMedia Laboratories, Mumbai, India) [18]. The enrichment culture was incubated in a microaerobic atmosphere generated by CampyGen (Oxoid, Basingstoke, UK) at 42°C for 24 hours. After incubation, two loopfuls of the enrichment culture were plated onto mCCDA. The mCCDA plates were incubated at 42°C for 48 hours in a micro-aerobic atmosphere. If no growth was observed, the plates were re-incubated under microaerobic conditions for an additional 24 hours, extending the total incubation time to 72 hours.

## Confirmation and species identification

Typical or suspected *Campylobacter* colonies were selected for further confirmation and species identification. The selected colonies were streaked onto a non-selective agar plate (Oxoid, Basingstoke, UK) containing 5% sheep blood and incubated at 37°C for 44 hours under microaerobic conditions, provided by CampyGen. DNA extraction was performed using a heat and snap-chilling method according to a previous study by Barakat et al. [19]. Two to three colonies of fresh bacterial growth were collected from the blood agar plate, suspended in nuclease-free deionized water, and then heated at 95°C for 10 minutes in a heat block. The samples were then cooled immediately and centrifuged at 13,000g for 5 minutes at room temperature. The supernatant, containing the template DNA, was carefully transferred to nuclease-free Eppendorf tubes and stored at -20°C until further use.

The Multiplex PCR (mPCR) method was applied for the confirmation of *Campylobacter* isolates, both at the genus and species level, as described in previous studies [20–23]. mPCR amplification was carried out in 25 µl reaction volumes using a Himedia Thermal Cycler. The reaction mixture contained the following components: 1 µl of template DNA, 25 µM of each primer, 1.5 mM MgCl₂, 1250 µM of each dNTP, 5 µM of the green fluorescent nucleic acid stain, 1 × reaction buffer, and 1 U of Taq DNA polymerase. The optimal mPCR conditions were as follows: initial denaturation at 96°C for 3 minutes, followed by 35 cycles of 96°C for 30 seconds, 57°C to 62°C for 90 seconds, and 72°C for 30 seconds. A final extension

step was carried out at 72°C for 10 minutes [24]. The PCR products were analyzed by electrophoresis on a 1.5% agarose gel in TAE running buffer (10X). Ethidium bromide (0.5 µg/ml) was used for staining, and DNA ladder along with 6x loading dye was applied for visualization during gel electrophoresis.

## Antimicrobial susceptibility testing

The antimicrobial susceptibility of isolated *Campylobacter* was tested using the disc diffusion method, following European Committee on Antimicrobial Susceptibility Testing (EUCAST) guidelines version 15 [25]. Three to four morphologically pure bacterial colonies were picked and suspended in sterile saline. The bacterial suspension was standardized by comparing it to a 0.5 McFarland standard. A loopful of the suspension was streaked onto Mueller-Hinton agar (HiMedia) supplemented with 5% sheep blood and evenly spread across the plate using a sterile cotton-tipped applicator. Antibiotic discs were placed on the plate using sterile forceps, spaced 20 mm apart. The plates were then incubated at 42°C for 24–48 hours under microaerobic conditions (5% $O_2$, 10% $CO_2$, and 85% $N_2$) generated using CampyGen. Inhibition zones were measured at 24 and 48 hours and interpreted according to EUCAST guidelines. The antibiotics were selected based on their availability and common use in Ethiopia, and belonged to five antibiotic classes: quinolones (nalidixic acid (30 µg), ciprofloxacin (5 µg), and norfloxacin (10 µg)), macrolides (erythromycin (15 µg)), aminoglycosides (streptomycin (10 µg) and gentamicin (10 µg)), phenicols (chloramphenicol (30 µg)), and tetracyclines (tetracycline (30 µg) and oxytetracycline (30 µg)). Multi-Drug Resistance (MDR) was defined as non-susceptibility to at least three antibiotic classes. Established breakpoints according to EUCAST guidelines were only available for ciprofloxacin, erythromycin, and tetracycline. Due to the absence of validated cut-off values for *Campylobacter* resistance to other antibiotics at the time of this study, only an inhibition zone of 0 mm was considered resistant.

## Data management and analysis

Data from field surveys and laboratory results were recorded using Microsoft Excel 2010 (Microsoft Corporation, USA). Analysis was performed using STATA version 18.0 software (StataCorp LLC, USA). Descriptive statistics were used to summarize the data. To examine the relationship between risk factors and the occurrence of *Campylobacter* in chickens and humans, univariable logistic regression analysis was conducted, with results expressed as crude odds ratios (COR) and 95% confidence intervals (CI). Variables with $P < 0.20$ in the univariable analysis were further included in the multivariable logistic regression analysis. Results are reported as adjusted odds ratios (AOR) with 95% CI. A forward stepwise approach was used for model development, and the goodness-of-fit of the final model was assessed using the Hosmer-Lemeshow test with 10 groups. Multicollinearity and interaction effects were checked. The model's predictive power was evaluated using the receiver operating characteristic (ROC) curve, and a P-value below 0.05 was considered as statistically significant.

## Results

### Description of the poultry farms

Of the 122 poultry farms visited in Debre Berhan town and surrounding areas, 63.9% were located in peri-urban, 22.2% in urban, and 13.9% in rural areas. The farming system comprised 47.5% smallholder farms (defined as those with 100 or fewer poultry, using local feed and basic management practices) and 52.5% commercial farms. Among the farms, 57.4% kept layers, 22.9% growers, 14.7% broilers, and 4.9% were mixed (layer and broiler). A significant proportion (72.9%) of farms lacked basic biosecurity measures, such as footbaths or changing shoes at the poultry house gate and disposed of dead poultry in garbage or open air near the poultry house. Additionally, 42.6% of farms did not restrict access to unauthorized personnel, 45.9% did not separate sick poultry, and 64.7% used window ventilation. Furthermore, 66.4% of farms kept other animals, including cattle, small ruminants, pets, and swine. Prophylactic use of antibiotics was reported in

53 (43.4%) of the poultry farms visited (S1 Table). Commonly used antibiotics reported by the respondents were oxytetracycline, Tylosin Tartrate (ASHTYL 20% Powder), and Enrofloxacin (Ashinero 10% Oral Solution). All the 53 farms mentioned oxytetracycline as a commonly used antibiotic.

## Occurrence of *Campylobacter*

*Campylobacter* species were detected in 46 (12.5%) of 366 samples, with the highest prevalence in poultry (19.6%), followed by human samples (13.1%) and poultry house floor samples (4.9%). Of the 46 *Campylobacter* isolates, 37 (80.4%) were identified as *C. jejuni* and 9 (19.6%) as *C. coli* (Table 1). *Campylobacter* species were found in at least one sample from 39 (32.0%) of the 122 poultry farms. Among these 39 farms, *Campylobacter* was found only in human stool on 12 (9.8%) farms, only in cloacal swabs on 18 (14.8%) farms, and only in poultry house floor samples on 3 (2.5%) farms. Additionally, *C. jejuni* was isolated from multiple sample types on 6 (4.9%) farms. Specifically, *C. jejuni* was detected in both human stool and cloacal swabs in 3 (2.5%) farms, in both cloacal swabs and poultry house floor samples in 2 (1.6%) farms, and in human stool, cloacal swab, and poultry house floor samples in 1 (0.8%) farm.

## Farm practices and risk factors for *Campylobacter* colonization in poultry

In univariable logistic regression (shown as Crude Odds Ratio; COR in Table 2), significantly higher *Campylobacter* prevalence was observed in poultry from farms that lacked footbaths or did not change footwear (COR = 5.1; 95% CI:1.1, 23; P = 0.035), disposed of poultry waste in the backyard (COR = 3.5; 95% CI:1.3, 9.3; P = 0.011), allowed unauthorized access to the poultry house (COR = 3.4; 95% CI:1.3, 8.8; P = 0.010), failed to separate sick poultry (COR = 2.9; 95% CI:1.1, 7.4; P = 0.026), did not practice all-in/all-out management (COR = 14.8; 95% CI: 4.9, 44.5; P = 0.000), and owned cattle (COR = 10.9; 95% CI: 2.3, 50.8; P = 0.002).

In multivariable logistic regression (shown as Adjusted Odds Ratio; AOR in Table 2), all-in/all-out management was identified as a protective factor against *Campylobacter* colonization in poultry. Poultry from farms that did not implement all-in/all-out management were 8.4 times (AOR = 8.4; 95% CI: 2.2, 32.4; P = 0.002) more likely to be colonized with *Campylobacter* compared to poultry from farms that practiced this management. Poultry from farms that owned cattle were 9.5 times (AOR = 9.5; 95% CI: 1.5, 58.2; P = 0.014) more likely to be colonized by *Campylobacter* compared to poultry from farms without cattle (Table 2).

## Human consumption habits and hygiene practices

Two-thirds (64.0%) of respondents reported consuming raw meat from ruminants, mainly beef. In univariable logistic regression (shown as COR in Table 3), significantly higher prevalence rates were found among individuals with no formal education (COR = 7.8; 95% CI: 1.5, 39.2; P = 0.012), farm attendants (COR = 10.7; 95% CI: 1.4, 84.5; P = 0.024), those who

**Table 1. Occurrence and distribution of *Campylobacter* species across sample types from poultry farms in and around Debre Berhan town, Ethiopia (n = 366).**

| Sample type | No. of samples | Positive | | | OR | |
|---|---|---|---|---|---|---|
| | | *C. jejuni* (%) | *C. coli* (%) | Total (%) | 95% CI | P-value |
| Cloacal swabs | 122 | 17 (13.9) | 7 (5.7) | 24 (19.6) | 4.7 (1.9-12.0) | 0.001 |
| Human stool | 122 | 14 (11.5) | 2 (1.6) | 16 (13.1) | 2.9 (1.1-7.7) | 0.031 |
| Poultry house floor sock samples | 122 | 6 (4.9) | 0 (0.0) | 6 (4.9) | 1 | – |
| Total | 366 | 37 (10.1) | 9 (2.4) | 46 (12.5) | – | – |

*OR, odds ratio; 95% CI, confidence interval at 95%.*

**Table 2.** Univariable and multivariable logistic regression analyses of risk factors associated with *Campylobacter* prevalence in poultry in and around Debre Berhan town (n = 122).

| Variables | No. of samples (%) | Positive (%) | Crude Odds Ratio | | Adjusted Odds Ratio | |
|---|---|---|---|---|---|---|
| | | | 95% CI | P-value | 95% CI | P-value |
| **Districts** | | | | | | |
| Debre Berhan town | 32 (26.2) | 7 (21.9) | 1.2 (0.4-3.2) | 0.715 | | |
| Debre Berhan town surroundings | 90 (73.8) | 17 (18.9) | 1 | | | |
| **Farm location** | | | | | | |
| Urban | 27 (22.2) | 6 (22.2) | 2.1 (0.4-12.1) | 0.388 | | |
| Peri-urban | 78 (63.9) | 16 (20.5) | 1.9 (0.4-9.3) | 0.411 | | |
| Rural | 17 (13.9) | 2 (11.7) | 1 | | | |
| **Farming system** | | | | | | |
| Small holder | 58 (47.5) | 10 (17.2) | 1 | | | |
| Commercial | 64 (52.5) | 14 (21.8) | 1.3 (0.5-3.3) | 0.521 | | |
| **Type of poultry** | | | | | | |
| Broiler | 18 (14.8) | 4 (22.2) | 1.3 (0.3-5.7) | 0.716 | | |
| Layer | 70 (57.4) | 13 (18.6) | 1.04 (0.3-3.2) | 0.934 | | |
| Mixed | 6 (4.9) | 2 (33.3) | 2.3 (0.3-16.2) | 0.403 | | |
| Grower | 28 (22.9) | 5 (17.8) | 1 | | | |
| **Flock size (average flock size = 373)** | | | | | | |
| Small (<20) | 25 (20.5) | 4 (16.0) | 1 | | | |
| Medium (21–99) | 29 (23.8) | 3 (10.3) | 0.6 (0.1-3.0) | 0.540 | | |
| Large (100–999) | 54 (44.2) | 14 (25.9) | 1.8 (0.5-6.3) | 0.332 | | |
| Very large (1000–4000) | 14 (11.5) | 3 (21.4) | 1.4 (0.3-7.5) | 0.673 | | |
| **Availability of ventilation** | | | | | | |
| Mechanical ventilation | 25 (20.5) | 6 (24.0) | 1 | | | |
| Window | 79 (64.8) | 14 (17.7) | 0.6 (0.23-2.0) | 0.489 | | |
| Under roof | 16 (13.1) | 4 (25.0) | 1.05 (0.2-4.5) | 0.942 | | |
| No ventilation | 2 (1.6) | 0 (0.0) | – | | | |
| **Disposal of poultry waste** | | | | | | |
| Backyard | 55 (45.1) | 17 (29.8) | 3.5 (1.3-9.3) | 0.011 | 1.3 (0.32-5.89) | 0.109 |
| Buried, burn or sell | 67 (54.9) | 7 (10.7) | 1 | | | |
| **Use footbaths or change the** footwear | | | | | | |
| Yes | 33 (27.1) | 2 (6.0) | 1 | | | |
| No | 89 (72.9) | 22 (24.7) | 5.1 (1.1-23.0) | 0.035 | 3.8 (0.62-23.6) | 0.148 |
| **Rodent and insect control** | | | | | | |
| Yes | 51 (41.8) | 9 (17.6) | 1 | | | |
| No | 71 (58.2) | 15 (21.1) | 1.2 (0.5-3.1) | 0.634 | | |
| **Drinker and feeder disinfection** | | | | | | |
| Yes | 52 (42.6) | 8 (15.4) | 1 | | | |
| No | 70 (57.4) | 16 (22.8) | 1.6 (0.9-5.9) | 0.307 | | |
| **Apply all-In/all-out management** | | | | | | |
| Yes | 83 (68.1) | 5 (6.0) | 1 | | | |
| No | 39 (31.9) | 19 (48.7) | 14.8 (4.9-44.5) | 0.000 | 8.4 (2.2-32.4) | 0.002 |
| **Restricted access to the flock** | | | | | | |
| Yes | 70 (57.4) | 8 (11.4) | 1 | | | |
| No | 52 (42.6) | 16 (30.7) | 3.4 (1.3-8.8) | 0.010 | 2.5 (0.6-10.7) | 0.210 |

*(Continued)*

**Table 2.** (Continued)

| Variables | No. of samples (%) | Positive (%) | Crude Odds Ratio | | Adjusted Odds Ratio | |
|---|---|---|---|---|---|---|
| | | | 95% CI | P-value | 95% CI | P-value |
| **Separate sick poultry in the farm** | | | | | | |
| Yes | 66 (54.1) | 8 (12.1) | 1 | | | |
| No | 56 (45.9) | 16 (28.6) | 2.9 (1.1-7.4) | 0.026 | 1.4 (0.4-5.5) | 0.627 |
| **Handling of dead poultry** | | | | | | |
| Throw to garbage | 89 (72.9) | 21 (23.6) | 3 (0.8-11.1) | 0.085 | 1.3(0.2-7.4) | 0.735 |
| Buried or burn | 33 (27.1) | 3 (9.1) | 1 | | | |
| **Type of Confinement** | | | | | | |
| Indoor part time | 48 (39.3) | 16 (33.3) | 1.4 (0.5-3.8) | 0.414 | | |
| Indoor exclusively | 70 (57.4) | 8 (29.6) | 1 | | | |
| Outdoor | 4 (3.3) | 0 (0.0) | – | | | |
| **Source of water** | | | | | | |
| Tap water | 74 (60.7) | 19 (25.6) | 2.8 (1.0-8.4) | 0.051 | 3.7 (0.7-18.5) | 0.109 |
| River | 46 (37.7) | 5 (10.8) | 1 | | | |
| Well | 2 (1.6) | 0 (0.0) | – | | | |
| **Presence of other animal species on the farm** | | | | | | |
| No | 41 (33.6) | 2 (4.8) | 1 | | | |
| Cattle | 50 (40.9) | 18 (36.0) | 10.9 (2.3-50.8) | 0.002 | 9.5 (1.5-58.2) | 0.014 |
| Pet (dog and cat) | 8 (6.6) | 1 (12.5) | 2.7 (0.22-35.0) | 0.428 | 9.9 (0.4-221) | 0.146 |
| Small ruminant | 15 (12.3) | 2 (13.3) | 3 (0.38-23.5) | 0.295 | 3.5 (0.2-44.5) | 0.333 |
| Swine | 8 (6.6) | 1 (12.5) | 2.7 (0.22-35.0) | 0.428 | 4.6 (0.2-139) | 0.372 |
| ***Campylobacter* positivity in human stool on the farm** | | | | | | |
| Positive | 16 (13.1) | 4 (25.0) | 1.4 (0.41-4.91) | 0.570 | | |
| Negative | 106 (86.9) | 20 (18.8) | 1 | | | |
| ***Campylobacter* contamination status of poultry house floors** | | | | | | |
| Positive | 6 (4.9) | 3 (50.0) | 4.5 (0.85-24.0) | 0.076 | 1.04 (0.1-11.0) | 0.975 |
| Negative | 116 (95.1) | 21 (18.1) | 1 | | | |

*Hosmer-Lemeshow chi ($x^2$) =13.36, P=0.100; ROC curve = 0.901 (90%).*

slaughtered animals at home (COR = 3.8; 95% CI: 1.0, 14.3; P = 0.043), individuals who did not use personal protective equipment (PPE) when handling poultry (COR = 7.7; 95% CI: 2.1, 28.9; P = 0.002), those who did not wash their hands with soap after contact with live animals (COR = 7.6; 95% CI: 2.3, 25.4; P = 0.001), those who disposed of poultry waste using a shovel (COR = 12.7; 95% CI: 1.4, 44.9; P = 0.023), and individuals unaware of zoonotic diseases (COR = 5.16; 95% CI: 1.1, 23.8; P = 0.036) (Table 3). However, in a multivariable logistic regression analysis (shown as AOR in Table 3), individuals who did not use PPE when handling poultry had 8.3 times (AOR = 8.3; 95% CI: 1.9, 36.5; P = 0.005) higher odds of contracting *Campylobacter*, and not washing hands after contact with live animals and cleaning barn increased the odds by 5.6 times (AOR = 5.6; 95% CI: 1.4, 21.8; P = 0.013). Consuming raw dairy products raised the odds by 5.5 times (AOR = 5.5; 95% CI:1.0, 29.9; P = 0.046), and allowing poultry in human sleeping and food preparation areas increased the odds by 6.3 times (AOR = 6.3; 95% CI: 1.3, 30.2; P = 0.020) (Table 3).

## Awareness of poultry farm workers

The majority of respondents, 75 (61.5%) out of 122, were unaware of zoonotic diseases, while the remaining 47 (38.5%) reported being aware. Among the 47 informed respondents, the zoonotic diseases mentioned included tuberculosis

**Table 3.** Univariable and multivariable logistic regression analyses of risk factors for *Campylobacter* infection in humans in and around Debre Berhan Town, Ethiopia (n = 122).

| Variables | No. of samples (%) | Positive (%) | Crude Odds Ratio | | Adjusted Odds Ratio | |
|---|---|---|---|---|---|---|
| | | | 95% CI | P-value | 95% CI | P-value |
| **Gender** | | | | | | |
| Male | 68 (55.7) | 10 (14.7) | 1.4 (0.4-4.1) | 0.469 | | |
| Female | 54 (44.3) | 6 (11.1) | 1 | | | |
| **Age (*Mean age: 27 years*)** | | | | | | |
| ≤20 years | 27 (22.1) | 3 (11.1) | 1 | | | |
| 21-30 years | 55 (45.1) | 7 (12.7) | 1.1 (0.3-4.9) | 0.834 | | |
| 31-50 years | 40 (32.8) | 6 (15.0) | 1.4 (0.3-6.2) | 0.648 | | |
| **Marital status** | | | | | | |
| Single | 51 (41.8) | 6(11.7) | 1 | | | |
| Married | 71 (58.2) | 10 (14.1) | 1.2 (0.4-3.6) | 0.708 | | |
| **Level of education** | | | | | | |
| No formal education | 32 (26.2) | 11 (34.4) | 7.8 (1.5-39.2) | 0.012 | | |
| Primary school | 32 (26.2) | 2 (6.2) | 1 | | | |
| High school | 38 (31.2) | 2 (5.2) | 0.8 (0.1-6.2) | 0.860 | | |
| Higher education | 20 (16.4) | 1 (5.0) | 0.8 (0.1-9.3) | 0.851 | | |
| **Occupation** | | | | | | |
| Farm attendant | 75 (61.5) | 15 (20.0) | 10.7 (1.4-84.5) | 0.024 | | |
| Farm manager | 44 (36.1) | 1 (2.3) | 1 | | | |
| Other[a] | 3 (2.4) | 0 (0.0) | – | | | |
| **Eat undercooked meat** | | | | | | |
| Yes | 78 (63.9) | 13 (16.7) | 2.7 (0.7-10.1) | 0.134 | | |
| No | 44 (36.1) | 3 (6.8) | 1 | | | |
| **Consume raw milk** | | | | | | |
| Yes | 71 (58.2) | 14 (20) | 6 (1.3-27.8) | 0.022 | 5.5 (1.0-29.9) | 0.046 |
| No | 51 (41.8) | 2 (3.9) | 1 | | | |
| **Poultry access to human sleeping and food preparation areas** | | | | | | |
| Yes | 77 (63.1) | 13 (16.9) | 2.8 (0.7-10.6) | 0.119 | 6.3 (1.3-30.2) | 0.020 |
| No | 45 (36.9) | 3 (6.7) | 1 | | | |
| **Slaughter domestic animals at their home** | | | | | | |
| Yes | 69 (56.6) | 13 (18.8) | 3.8 (1.0-14.3) | 0.043 | | |
| No | 53 (43.4) | 3 (6.9) | 1 | | | |
| **Use of PPE[b] during contact with poultry** | | | | | | |
| Yes | 71 (58.2) | 3 (4.2) | 1 | | | |
| No | 51 (41.8) | 13 (25.5) | 7.7 (2.1-28.9) | 0.002 | 8.3 (1.9-36.5) | 0.005 |
| **Wash hands with soap after handling live poultry and cleaning barn** | | | | | | |
| Yes | 80 (65.6) | 4 (5.0) | 1 | | | |
| No | 42 (34.4) | 12 (28.6) | 7.6 (2.3-25.4) | 0.001 | 5.6 (1.4-21.8) | 0.013 |
| **How do you dispose of poultry waste[c]?** | | | | | | |
| Using gloves | 35 (28.7) | 1 (2.8) | 1 | | | |
| By bare hand | 65 (53.3) | 9 (13.8) | 5.4 (0.7-45.0) | 0.115 | | |
| Using shovel | 22 (18.0) | 6 (27.3) | 12.7 (1.4-44.9) | 0.023 | | |
| **Awareness of zoonotic diseases** | | | | | | |
| Yes | 47 (38.5) | 2 (4.2) | 1 | | | |
| No | 75 (61.5) | 14 (18.7) | 5.16 (1.1-23.8) | 0.036 | | |

*(Continued)*

**Table 3.** (Continued)

| Variables | No. of samples (%) | Positive (%) | Crude Odds Ratio | | Adjusted Odds Ratio | |
|---|---|---|---|---|---|---|
| | | | 95% CI | P-value | 95% CI | P-value |
| *Campylobacter*-positive poultry on the farm | | | | | | |
| Positive | 24 (19.7) | 4 (16.7) | 1.4 (0.4-4.9) | 0.567 | | |
| Negative | 98 (80.3) | 12 (12.2) | 1 | | | |
| *Campylobacter* contamination status of poultry house floors | | | | | | |
| Positive | 6 (4.9) | 1 (16.7) | 1.3 (0.1-12.3) | 0.792 | | |
| Negative | 116 (95.1) | 15 (12.9) | 1 | | | |

[a]Teacher or animal health professional or bankers; [b]PPE, personal protective equipment; wastes[c], manure, aborted fetus; Hosmer-Lemeshow chi ($\chi^2$) = 1.01, P=0.994; ROC curve=0.872 (87%).

(n=13), cholera (n=10), salmonellosis (n=6), anthrax (n=2), and both tuberculosis and cholera (n=5). Eleven respondents simply mentioned 'bacteria' as a zoonotic disease. None of the respondents were aware of *Campylobacter* or recognized it as a zoonotic pathogen. Only 5 (4.1%) respondents were familiar with the One Health concept. Regarding attitudes toward zoonotic disease transmission, 57 (46.7%) believed that diarrhea-causing agents could be transmitted from animals to humans. The routes of transmission identified by these 57 respondents were: consumption of contaminated raw meat (n=13) contaminated milk and meat (n=12), contaminated meat and water (n=8), contaminated raw milk (n=7), contact with infected animals (n=9), exposure to contaminated water (n=4), and respondents who were unsure or said I don't know (n=4).

## Antimicrobial resistance profiles

All 46 *Campylobacter* isolates exhibited resistance to at least one antimicrobial. Overall, the highest resistance was observed to tetracycline (89.1%), followed by ciprofloxacin (69.5%) and erythromycin (63.0%). Among the 37 *C. jejuni* isolates, 92.0% were resistant to tetracycline, 70.2% to ciprofloxacin, and 64.8% to erythromycin. For the 9 *C. coli* isolates, resistance rates were 77.8% to tetracycline, 66.7% to ciprofloxacin, and 55.5% to erythromycin. By considering only no inhibition of growth (0 mm), resistance was also observed for other antimicrobials: 28.2% of isolates were resistant to norfloxacin (32.4% *C. jejuni* and 11.1% *C. coli*), 30.4% to streptomycin (29.7% *C. jejuni* and 33.3% *C. coli*), 13.5% to chloramphenicol (*C. jejuni*), 21.7% to gentamicin (24.3% *C. jejuni* and 11.1% *C. coli*), 19.5% to nalidixic acid (18.9% *C. jejuni* and 22.2% *C. coli*), 73.9% to oxytetracycline (78.3% *C. jejuni* and 55.5% *C. coli*) (Table 4).

A total of 32 (69.5%) out of 46 *Campylobacter* isolates were MDR. This included 28 (75.7%) of 37 *C. jejuni* isolates and 4 (44.4%) of 9 *C. coli* isolates. Among the 28 MDR *C. jejuni* isolates, 14 (82.3%) of 17 were from poultry, 9 (64.3%) of 14 from humans, and 5 (83.3%) of 6 from poultry house floors. The four MDR *C. coli* isolates were obtained from three poultry samples and one human sample. The most frequently observed MDR pattern involved resistance to quinolones, macrolides, and tetracyclines (Table 4). All co-occurring *C. jejuni* isolated from the same farms were MDR. Similar resistance patterns were identified across all sample types on five of the six farms. On the remaining farm, however, resistance patterns differed between isolates. The cloacal swab isolates showed resistance to ciprofloxacin, erythromycin, tetracycline, nalidixic acid, streptomycin, chloramphenicol, gentamicin, and oxytetracycline. In contrast, the human isolates was resistant to ciprofloxacin, erythromycin, tetracycline, streptomycin, chloramphenicol, gentamicin, and oxytetracycline.

**Table 4. Distribution of antimicrobial resistance among 37 *C. jejuni* and 9 *C. coli* isolates obtained from human stool, cloacal swabs, and poultry house floors samples in and around Debre Berhan, Ethiopia. Grey-highlighted rows indicate MDR isolates, defined as resistant to at least three different antibiotic classes.**

| *C. jejuni* resistance | Human (n = 14) | Cloacal (n = 17) | Poultry house floor sock samples (n = 6) | Total |
|---|---|---|---|---|
| Cip + Ery + Tet | 3 | – | – | 3 |
| Cip + Ery + Tet + Oxy* | 1 | 2 | 1 | 4 |
| Cip + Ery + Tet + Oxy*+Nor* | 1 | 2 | 1 | 4 |
| Cip + Ery + Tet + Oxy*+Nor* + Str* | 1 | 1 | – | 2 |
| Cip + Ery + Tet + Oxy*+Nor* + Str* + Nal* | 1 | 1 | 1 | 3 |
| Cip + Ery + Tet + Oxy*+Nor* + Gen* + Nal* | – | 1 | 1 | 2 |
| Cip + Ery + Tet + Oxy* + Gen* | – | 3 | – | 3 |
| Cip + Tet + Oxy* + Str* + Gen* | 1 | – | – | 1 |
| Cip + Ery + Tet + Oxy* + Str* + Gen* + Chl* | 1 | – | – | 1 |
| Cip + Ery + Tet + Oxy*+Nor* + Gen* + Str* + Chl* | – | 1 | – | 1 |
| Ery + Tet + Oxy* + Str* | – | 2 | – | 2 |
| Ery + Tet + Oxy* + Chl* | – | 1 | 1 | 2 |
| Ery + Tet + Oxy* | – | – | 1 | 1 |
| Ery + Tet | 2 | – | – | 2 |
| Cip + Tet + Oxy* | 1 | – | – | 1 |
| Tet + Oxy* | – | 2 | – | 2 |
| Ery | 2 | – | – | 2 |
| Str* | – | 1 | – | 1 |
| *C. coli* resistance | Human (n = 2) | Cloacal (n = 7) | Poultry house floor sock samples (n = 0) | |
| Cip + Ery + Tet + Oxy* | 1 | 1 | – | 2 |
| Cip + Ery + Tet + Oxy* + Str* + Gen* + Chl* | – | 1 | – | 1 |
| Cip + Ery + Tet + Oxy* + Str*+Nor* | – | 1 | – | 1 |
| Cip + Nal* | – | 1 | – | 1 |
| Cip + Nal* + Tet + Oxy* | – | 1 | – | 1 |
| Str* | – | 1 | – | 1 |
| Ery | 1 | – | – | 1 |
| Gen* | – | 1 | – | 1 |

*\* No validated cut-off value is available, resistance is defined as absence of growth inhibition (0 mm). Abbreviations: Cip, Ciprofloxacin; Ery, Erythromycin; Tet, Tetracycline; Oxy, Oxytetracycline; Nor, Norfloxacin; Str, Streptomycin; Nal, Nalidixic acid; Gen, Gentamicin and Chl, Chloramphenicol.*

## Discussion

The occurrence of *Campylobacter* in this study aligns with previous research conducted in Ethiopia [12,26,27], Kenya [28], Sub-Saharan Africa [29], and Cambodia [30]. However, our findings indicate a lower prevalence compared to earlier studies in Ethiopia [31–33] and Tanzania [34]. This variation may be attributed to differences in geography, detection methods, sampling protocols, animal management, and sanitation practices.

In this study, *Campylobacter* was detected in 4.9% of poultry house floor samples, highlighting fecal contamination as a key factor in the spread of *Campylobacter*. A notably higher prevalence of *Campylobacter* (18.4%) has been reported in a previous study conducted in Ethiopia [35]. This discrepancy may reflect a genuine difference in prevalence rates or could be attributed to variations in detection methodologies. Specifically, the aforementioned study utilized direct polymerase chain reaction (PCR) techniques on environmental samples, whereas our investigation employed culture methods with enrichment using buffered peptone water without the addition of antibiotic supplement to inhibit microorganisms such as

coliform bacteria, yeast and molds. The isolates were subsequently confirmed by PCR. This methodological divergence may account for the lower prevalence observed in our study.

*Campylobacter jejuni* was the dominant species isolated in this study, consistent to previous findings [29–34], highlighting its critical role in human campylobacteriosis [10]. The co-occurrence of *C. jejuni* across various sample types within the same farm across poultry, humans, and the environment, suggests complex transmission dynamics at the farm level. This potential intra-farm transmission is further supported by the observation of similar AMR patterns across these isolates, all of which were MDR, indicating a potential transmission cycle within the farm environment.

According to the World Organization for Animal Health (WOAH) [36], the all-in/all-out management system is a biosecurity practice where poultry of the same age are introduced and removed from a production unit simultaneously. The facility is then cleaned and left empty before restocking. This approach helps prevent disease transmission and enhances production efficiency. Our study found that the all-in/all-out management system significantly reduces *Campylobacter* colonization in poultry. The absence of this system in some farms highlights a critical biosecurity gap that contributes to the spread of *Campylobacter*. The all-in/all-out approach minimizes cross-contamination between flocks and facilitates thorough cleaning between batches, thus reducing bacterial loads. Inadequate cleaning, partial depopulation, and poor disinfection practices between batches have been identified as risk factors for *Campylobacter* colonization [37]. Additionally, farms that housed both poultry and cattle were found to have higher odds of having *Campylobacter*-positive poultry, suggesting cross-species transmission. Previous studies have consistently identified other animals, particularly cattle, on or near farms as significant risk factors [6,38], likely contributing to the maintenance of *Campylobacter* between flocks [37]. Therefore, other domestic animal species could play a role in *Campylobacter* transmission to humans, especially in countries such as Ethiopia, where mixed livestock farming is a common practice.

This study also identified raw milk consumption as a risk factor for human *Campylobacter* infection, consistent with previous findings [39,40]. In addition, allowing poultry access to human sleeping and food preparation areas significantly increased the likelihood of infection, similar to findings from Cambodia [30]. Poor hygiene practices, such as not washing hands after handling poultry or cleaning poultry barns, and not using PPE, were also associated with higher infection rates. These results reinforce the importance of hygiene as a major risk factor for *Campylobacter* transmission, as highlighted in previous studies [26,41].

A significant knowledge gap was observed among farm workers regarding zoonotic diseases and the One Health concept, with all workers unaware that *Campylobacter* is zoonotic. This aligns with findings from Abunna et al. [42], who also identified limited awareness of zoonotic risks as a barrier to improving public health outcomes. Our univariable analysis showed that *Campylobacter* prevalence was higher in individuals unaware of its zoonotic nature, which may be associated with their lack of awareness. These findings underscore the need for targeted educational interventions to raise awareness about zoonotic risks and promote One Health strategies to reduce risks to both animal and human health [43].

Antimicrobial resistance was a major concern in this study. All *Campylobacter* isolates showed resistance to at least one antimicrobial agent, with notably high levels of resistance to tetracycline, erythromycin, and ciprofloxacin. These findings are consistent with previous studies from Ethiopia [12,26], other Sub-Saharan African countries [29] and globally [8]. Such high resistance is likely driven by the widespread use of antibiotics as prophylactics, with nearly half of the poultry farms in the study area relying on them. Tetracyclines, especially oxytetracycline, have been widely used in Ethiopian animal production for many years. Improper antimicrobial use in poultry farming, coupled with insufficient regulatory oversight, has contributed to the emergence of resistant bacterial strains [12,44,45]. The long-term, uncontrolled use of fluoroquinolones and tetracycline, coupled with poor regulatory oversight, has facilitated the spread of resistance [11].

The high MDR rate observed in this study represents a serious health issue, as most isolates were resistant to commonly used antibiotics. The true MDR rate was most likely much higher, as resistance was defined as a complete absence of growth inhibition (0 mm) for certain antibiotics. Similar findings have been reported from Ethiopia [26,46,47], Kenya [28], and Bangladesh [48]. The dominant MDR pattern includes resistance to quinolones, macrolides, and tetracyclines,

reflecting global trends of high resistance, particularly to tetracycline and ciprofloxacin [49]. The co-occurrence of MDR isolates across multiple sample types on the farms highlights the emergence and local transmission of resistant *Campylobacter*. Bidirectional transmission between humans and animals, coupled with widespread prophylactic and therapeutic antimicrobial use, leads to continuous antibiotic pressure, facilitating the increase and spread of MDR *Campylobacter* [47,50].

While this study provides valuable insights into *Campylobacter* occurrence and risk factors across human, poultry, and environmental samples, a few limitations should be considered. The cross-sectional design, and absence of genetic analysis, limited our ability to assess transmission dynamics between humans and animals. The small subgroup sizes also limited our ability to assess AMR patterns by farm type. Although the sample size in each One Health domain was relatively small, key risk factors were identified. Culture-based methods successfully detected *C. jejuni* and *C. coli* but may have missed less common or emerging species. Environmental sampling was limited to poultry floor only, and detection methods without antibiotic supplements, along with the inability to detect non-viable *Campylobacter*, likely contributed to the observed lower prevalence. Additionally, some delays between sample collection and processing, as well as the lack of EUCAST breakpoints for some antibiotics, may have affected detection and resistance profiling. Future studies incorporating molecular techniques and expanded sampling would enhance the detection and characterization of *Campylobacter* and strengthen the understanding of environmental transmission pathways.

## Conclusion

In conclusion, this study highlights the widespread occurrence of *Campylobacter* in poultry and humans, along with concerning levels of antimicrobial resistance. Poor biosecurity and management practices contributed to *Campylobacter* colonization in poultry, while human infections were primarily associated with raw milk consumption and inadequate hygiene practices on poultry farms. The observed knowledge gap among farm workers underscores the need for targeted education interventions. These findings emphasize the value of a One Health approach. Future interventions in Ethiopia should focus on strengthening farm-level biosecurity, improving hygiene practices, and implementing antimicrobial stewardship programs to reduce risks to both animal and human health.

## Supporting information

**S1 Table. Description of poultry farms in Debre Berhan and surrounding areas, and occurrence of *Campylobacter* at the farm level.** A farm was considered Campylobacter-positive if the organism was isolated from at least one of the collected samples (human, poultry, or environmental) from that farm.
(DOCX)

## Acknowledgments

We acknowledge Dr. Wondwesen Azemeraw for assisting with data collection and Dr. Mequanint Addisu for his support with laboratory work. We also appreciate the poultry farm owners, district authorities, veterinarians, and field workers who facilitated and supported the fieldwork.

## Author contributions

**Conceptualization:** Fikre Zeru, Haileeyesus Adamu, Tesfaye Sisay Tessema, Ingrid Hansson, Sofia Boqvist.

**Data curation:** Fikre Zeru, Haileeyesus Adamu, Tesfaye Sisay Tessema, Ingrid Hansson, Sofia Boqvist.

**Formal analysis:** Fikre Zeru, Haileeyesus Adamu, Yohannes Hagos Woldearegay, Tesfaye Sisay Tessema, Ingrid Hansson, Sofia Boqvist.

**Funding acquisition:** Fikre Zeru, Haileeyesus Adamu, Tesfaye Sisay Tessema, Ingrid Hansson, Sofia Boqvist.

**Investigation:** Fikre Zeru, Haileeyesus Adamu, Tesfaye Sisay Tessema, Ingrid Hansson, Sofia Boqvist.

**Methodology:** Fikre Zeru, Haileeyesus Adamu, Yohannes Hagos Woldearegay, Tesfaye Sisay Tessema, Ingrid Hansson, Sofia Boqvist.

**Project administration:** Fikre Zeru, Haileeyesus Adamu, Tesfaye Sisay Tessema, Ingrid Hansson, Sofia Boqvist.

**Resources:** Fikre Zeru, Haileeyesus Adamu, Tesfaye Sisay Tessema, Ingrid Hansson, Sofia Boqvist.

**Software:** Fikre Zeru, Yohannes Hagos Woldearegay.

**Supervision:** Fikre Zeru, Haileeyesus Adamu, Tesfaye Sisay Tessema, Ingrid Hansson, Sofia Boqvist.

**Validation:** Fikre Zeru, Haileeyesus Adamu, Tesfaye Sisay Tessema, Ingrid Hansson, Sofia Boqvist.

**Visualization:** Fikre Zeru, Haileeyesus Adamu, Tesfaye Sisay Tessema, Ingrid Hansson, Sofia Boqvist.

**Writing – original draft:** Fikre Zeru.

**Writing – review & editing:** Fikre Zeru, Haileeyesus Adamu, Yohannes Hagos Woldearegay, Tesfaye Sisay Tessema, Ingrid Hansson, Sofia Boqvist.

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
