## [Decision Letter · Decision Letter 0]

3 Apr 2025

PNTD-D-25-00226Occurrence, Risk factors and Antimicrobial resistance of Campylobacter from Poultry and Humans in Central Ethiopia: A One Health ApproachPLOS Neglected Tropical Diseases Dear Dr. Kflewahd, Thank you for submitting your manuscript to PLOS Neglected Tropical Diseases. After careful consideration, we feel that it has merit but does not fully meet PLOS Neglected Tropical Diseases's publication criteria as it currently stands. Therefore, we invite you to submit a revised version of the manuscript that addresses the points raised during the review process. Please submit your revised manuscript within 30 days Jun 02 2025 11:59PM. If you will need more time than this to complete your revisions, please reply to this message or contact the journal office at plosntds@plos.org.  Please include the following items when submitting your revised manuscript: * A rebuttal letter that responds to each point raised by the editor and reviewer(s). You should upload this letter as a separate file labeled 'Response to Reviewers '. This file does not need to include responses to any formatting updates and technical items listed in the 'Journal Requirements' section below. * A marked-up copy of your manuscript that highlights changes made to the original version. You should upload this as a separate file labeled 'Revised Manuscript with Track Changes '. * An unmarked version of your revised paper without tracked changes. You should upload this as a separate file labeled 'Manuscript '. If you would like to make changes to your financial disclosure, competing interests statement, or data availability statement, please make these updates within the submission form at the time of resubmission. Guidelines for resubmitting your figure files are available below the reviewer comments at the end of this letter. We look forward to receiving your revised manuscript. Kind regards, Ben PascoeAcademic EditorPLOS Neglected Tropical Diseases Georgios PappasSection EditorPLOS Neglected Tropical Diseases

Shaden Kamhawi

co-Editor-in-Chief

Paul Brindley

co-Editor-in-Chief

**Additional Editor Comments:**  Your manuscript is promising and relevant. Revisions are needed to clarify sampling methods for both poultry and human subjects, ensure consistent terminology, and detail inclusion criteria. Please enhance data presentation with supplementary tables, verify numerical accuracy, and improve figure quality. Address study limitations, specify the EUCAST version used, and clarify questionnaire administration. Overall, your study’s design and public health relevance are commendable, and these refinements will strengthen the manuscript for publication.**Journal Requirements:** 

- ® on page: 12 and 13

- TM on page: 9.

4) Please amend your detailed Financial Disclosure statement. This is published with the article. It must therefore be completed in full sentences and contain the exact wording you wish to be published.

2) State what role the funders took in the study. If the funders had no role in your study, please state: "The funders had no role in study design, data collection and analysis, decision to publish, or preparation of the manuscript.".

**Reviewers' comments:**  Reviewer's Responses to Questions

**Key Review Criteria Required for Acceptance?**

**Methods**

-Are the objectives of the study clearly articulated with a clear testable hypothesis stated?

-Is the study design appropriate to address the stated objectives?

-Is the population clearly described and appropriate for the hypothesis being tested?

-Is the sample size sufficient to ensure adequate power to address the hypothesis being tested?

-Were correct statistical analysis used to support conclusions?

-Are there concerns about ethical or regulatory requirements being met?

Reviewer #1: - The study was based in central Ethiopia. Figure 1 shows a clear map of the study area.

- It is clearly noted that the study is a cross-sectional approach of the population during March 2021 to March 2022 on selected poultry farms. The reasoning behind the study districts is mentioned due to the high concentration of these poultry farms. Overall, 122 poultry farms were used in the study to ensure statistical power and reliability. The author's have noted that professionals in the field were enlisted to help identify suitable poultry farms for investigation. The methods for choosing the poultry farms for the study are very clear, with calculations of sample size performed using a public software tool for epidemiology (CDC).

- Could the authors state how the chicken samples were randomly chosen (lines 143 and 175) and whether the number of samples was consistent across the different poultry houses. Was it similar to the lottery method used for the human samples? It would also be good for the authors to expand how this lottery method was performed for choosing the human samples when multiple people were present.

- Could the authors better link what they mean by 'sampled from all compartments of the poultry house floor' (lines 144-145) to the explanation on lines 178-183. I was able to gain better understanding of the approach after reading the later description. Was this kept consistent in terms of area and number of samples for each poultry house?

- The author's clearly state that the structured questionnaire had open, closed, and semi-closed formatted questions (lines 157-158). The questionnaire was performed face-to-face using the local language, meaning that it was more likely that there would have been better response than a written questionnaire. The author's state that the questionnaire was pre-tested on ten farms to improve its clarity. Could the authors mention what ten farms these were and why/how they were chosen. How many improved questionnaires were then performed subsequently?

- Could the authors be specific/state whether there was a specific amount of hours for the person to provide a sample in (line 173).

- Lines 180-181: is it possible for the authors to be more specific about walking through all areas of the farm floor multiple times. Were the number of times recorded and were they consistent for each farm? How did you ensure that they did not miss any areas?

- Authors are overall very clear on how farm samples were stored before shipping and processing, as well as the isolation/identification of Campylobacter from the samples.

- Line 235: guideline would read better as plural (guidelines)

- Line 236: what do the authors mean by 'normal'? I think this word could be omitted, as I assume they just mean sterile saline.

- The authors describe well how the disc diffusion assays were performed, also noting that the discs were 20mm apart. This would have helped ensure that there was enough room for larger zones of clearing.

- Line 240: could the authors state specifically what the microaerobic conditions were ie the percentage of oxygen present.

- The authors mention the lack of EUCAST breakpoints for some antibiotics and that 0mm has been used as a cut-off for resistance. This is a conservative value, however, it is understandable why this was chosen in this instance. There are still no EUCASE breakpoints for the other antibiotics at the time of review.

- Methods overall clear to follow and well-written with only some minor adjustments required.

- Ethical considerations and approval, including the approval number, are clearly stated at the end of the methods.

Reviewer #2: Objectives were clearly stated and appropriate methodology used to address them, however there were some gaps in describing the study population and clarification needed in outlining the sampling locations. No concerns regarding ethical or regulatory requirements. The following outlines where more detail or clarification is needed:

• Clarification throughout that there were three study locations, rather than initially naming one then noting more further on in the text. Please specify which of the three locations are classified as rural/urban.

• Line 140, unclear whether the inclusion criteria considered the size of the farm, urban/rural location or whether it was a commercial farm or small holding- please include if they were.

• Line 143, was there any effort to capture information from those who worked on the farm and lived elsewhere as well as those who lived and worked on the farm? If so at the moment this is not clear.

• Consistency in how the poultry house floor/ boot/sock samples are referred to throughout would help the reader follow the narrative better. (including in Fig 2).

• Line 155, was the epi data collected from the same individual who provided the human sample? Please clarify

• Line 162, clarification needed on epidemiological data collection i.e. did the same person interview all of the individuals included in the study and was a translator used?

• Line 164, please specify whether questionnaire responses from the 10 farms the questionnaire was tested on were included in the analysis.

• Line 180, as noted above, please ensure clarity on where the boot sock samples were taken- was it just in the poultry houses or throughout the farm as this line implies the latter and line 144 states ‘all compartments of the poultry house’

• Line 191, please update heading, this section only refers to culture not identification.

• Line 260, please add which direction was used for the step-wise model development, either forward or backward.

Reviewer #3: Authors of the study used the right methodology (sampling, microbiology analysis and statistic analysis)to acheived their objectives.

**Results**

-Does the analysis presented match the analysis plan?

-Are the results clearly and completely presented?

-Are the figures (Tables, Images) of sufficient quality for clarity?

Reviewer #1: - In the description of the poultry farms, it would be good to see a supplementary table for this. It would make it clearer to see which farms used antibiotics, had other animals, were in specific areas etc. This would also enable better links to be made between the conditions of the poultry farms and the Campylobacter isolated/identified.

- Line 289: could the authors break down the percentage of oxytetracycline used on the farms out of the total 43.4% of antibiotics used.

- Lines 297-299: very good noting of the Campylobacter spp. detected and break down of the sample percentages.

- Line 301: When it states 'on these farms' is it referring to the 39/122 poultry farms in the sentence before? Please make this a little more explicit.

- Lines 304-306: you could add some percentages to these to supplement the raw numbers.

- Table 1 is a good addition to the text and supplements the results well

- Lines 316-321: Do the authors have any statistics for the individual remarks? Line 322 states about the AOR/CI for an all-in/all-out approach, but it would be good to see how it breaks down individually, too, if possible. This way it will show us whether there is lack of one (or more) specific approach that contributes to higher Campylobacter prevalence.

- Table 3 should say Ratio not Ration (line 351)

- Table 3: Can the authors explain why there are only AORs given for 4 of the categories and not other categories?

- How many respondents were there in total? The authors mention that 61.5% of respondents were unaware of zoonotic diseases (line 30). It would be good for them to add how many this was as a raw number as well.

- What is meant by ‘award respondents’? Line 361

- Lines 385-386: a little hard to read at first. Please could the authors reword this for clarity. This could possibly be split into two separate sentences: 1) How many Campylobacter isolates were MDR and 2) How many were part of each species.

Lines 390-391 also slightly hard to understand. These sentences could also be broken up a little or re-worded. ‘All co-occurring C. jejuni strains on the same farms were MDR’ is fine as a stand-alone sentence. The 2nd sentence could be phrased such as ‘similar resistance patterns were observed across all sample types on five of six farms’. Having the original sentence of ‘except for isolates from one farm’ is a little confusing at the end.

- Could the authors break table 4 down into resistance to all the singular antibiotics only as a separate table. This can then address the overall antibiotic resistance of the isolates, which can be referred to in-text, followed by MDR patterns of the combined antibiotics. Please can the authors add a 0 (or similar) to cells where there are no applicable isolates. The table currently looks unfinished without this.

Reviewer #2: Results presented are consistent with the approach outlined in the analysis plan but need reviewing to ensure accuracy of reported findings. On two occasions summarised in the text are not included as a table and p-values are not reported alongside COR/AORs. Tables and figures included are clearly laid out and look good.

The following outlines where more detail or clarification is needed:

• Lines 280 to 284, please ensure percentages quoted add up to 100%, in these lines they only total 99.9% each time.

• Data presented in lines 280 to 290 should be included as a table.

• Line 299, please check calculation of percentage of isolates which were C.coli, 9/46=19.6% not 24.3%.

• Line 316 to 319, reference table 2 and include CORs in the text so evidence of the significant results are clear, would also be beneficial to indicate where the strongest associations are- e.g. cattle/all-in-all-out management.

• Lines 322 and 325, please include p-value with the AOR.

• Lines 338 to 342, please include CORs in the text so evidence of the significant results are clear.

• Lines 344 to 348, please include p-value with the AOR.

• Lines 360 to 370, data presented in the text should be summarised in a table too, might be beneficial to include as a supplementary.

• Line 391, please explain the resistance pattern from isolates on the one farm which had different AMR

Reviewer #3: They used the right statistical analysis to identify to risk factors associated to antimicrobial resistance of Campylobacter from Poultry and Humans in Central Ethiopia. Results have been clearly presented, tables are sifficient quality, but the quality of image of the map need to improve.

**Conclusions**

-Are the conclusions supported by the data presented?

-Are the limitations of analysis clearly described?

-Do the authors discuss how these data can be helpful to advance our understanding of the topic under study?

-Is public health relevance addressed?

Reviewer #1: - Campylobacter jejuni is noted as critical in human infection and links to Campylobacter in other studies, including Ethiopia, has been noted and discrepancies addressed. There were also links made between higher Campylobacter incidence in farms where there is both poultry and cattle, which corroborates that of previously published articles. The authors have noted that this could be due to horizontal transmission. They highlight domestic animals as a source of Campylobacter transmission to humans.

- The authors clearly link a lot of Campylobacter risk factors identified in the study to other global studies. For example, drinking of raw milk, poor hygiene/lack of PPE, and allowing poultry to freely roam in food preparation areas. They have highlighted the requirement for education of farm workers around zoonotic diseases and the One Health concept. They note in the conclusions about future interventions within Ethiopia which may contribute to reducing Campylobacter risks.

- The discussion mentions about antimicrobial resistance being a major concern in the study, with all Campylobacter being resistant to at least one antibiotic. The authors have done well to link the findings from other studies both within Ethiopia and globally. The authors could also mention the limitations of the methods/results due to the lack of EUCAST break points for some antibiotics. Alternative methods such as minimum inhibitory concentrations and E-test strips could be mentioned but noted that these are either not time-effective or can be costly, and thus the disc diffusion assay was still the best choice for this study.

- I would like to see the authors address the limitations of their study a little clearer within the discussion. Whilst they make some very valid points and links to previous literature, there are potentially some limitations within the methods and results around the sampling of the poultry and the houses.

Reviewer #2: Conclusions drawn from the findings presented are supported and authors have clearly demonstrated how this study adds to the current evidence, which is somewhat limited. Novel One Health approach taken is very relevant and provides a good framework for future studies to investigate this topic in more detail.Findings are clearly linked to potential public health actions which can be taken to reduce human/animal health risks.

Reviewer #3: The conclusion supporte presented data. No limitation of analysis has been mentioned. The public health relevance of data has been addressed.

**Editorial and Data Presentation Modifications?**

Reviewer #1: Minor revisions with some expansions clarifying the comments above

Reviewer #2: This was a well written manuscript with relevant references included and only a few typos. I have outlined below where either further clarification or expansion on a point to provide more context would be better:

• Typos on line 103 (also), in Figure 1 (‘town’ in bottom panel), 361 (aware)

• Line 76, these references are for northern European countries only, please make this clear in the text or include references from a more directly comparable location

• Line 79- text here is a direct quote, but not indicated- please make this clear

• Line 92, more information on the context of the AMR rates for Campylobacter in Ethiopia would be helpful- were the samples taken from humans/poultry/livestock, when and was there a trend over time? Give the range quoted is 0% to 100% more information is needed to understand how applicable to the study and also may be more important to stress that estimates are variable/little is known

• Line 104, would suggest that this sentence is aligned with the abstract to also include the environment.

• Clarification on ‘all-in-all-out management’ meaning

• Line 391, It would be very interesting to know whether there was any pattern in resistance by type of farm (commercial vs small holding) or location (e.g. urban vs rural).

• Line 469 to 470- you note nearly half the farms in the study area reply on antibiotic use a prophylaxis, did you collect information on antibiotic use for each farm recruited to the study, if not this should be noted as a limitation

Reviewer #3: (No Response)

**Summary and General Comments**

Reviewer #1: I feel that this is a strong submission of Campylobacter epidemiology in Ethiopia. It is of particular interest as Campylobacter is under sampled in this area, so this paper will contribute well to the research field and the PLOS Neglected Tropical Diseases journal.

If the authors can address my minor comments and recommendations above, I believe this will be worthy of publication.

Reviewer #2: I really enjoyed reading this manuscript and I think it's a brilliant study which will not only add to the international evidence base but also provides a design which could easily be replicated in other countries to offer further insights. This work further highlights the importance of a One Health approach and was a well designed study. However, the manuscript would benefit significantly from a few revisions to ensure accuracy, clarity and also that all data presented in the text are available to the reader alongside for their consideration.

Reviewer #3: Campylobacter is a major cause au diarrheal diseases worldwide, and results you presented have a significance importance in public health.

I appreciated the effort you made to discussion yours results.

What version of EUCAST did you used to interpret the antimicrobial suseptibility test regarding oxytetracycline, streptomycin, nalidixic acid?

Can you pleased summarised title of tables you used in the results part? They are too longs.

You can now used legend to explain what you want.

Tiltle of figure 1 can be improve and be precise.

PLOS authors have the option to publish the peer review history of their article (what does this mean? ). If published, this will include your full peer review and any attached files.

**Do you want your identity to be public for this peer review?** For information about this choice, including consent withdrawal, please see our Privacy Policy .

Reviewer #1: **Yes: ** Kasia M Parfitt

Reviewer #2: No

Reviewer #3: No

---

## [Editor Report · Decision Letter 1]

22 Jun 2025

PNTD-D-25-00226R1Occurrence, Risk factors and Antimicrobial resistance of Campylobacter from Poultry and Humans in Central Ethiopia: A One Health ApproachPLOS Neglected Tropical Diseases Dear Dr. Kflewahd, Thank you for submitting your manuscript to PLOS Neglected Tropical Diseases. After careful consideration, we feel that it has merit but does not fully meet PLOS Neglected Tropical Diseases's publication criteria as it currently stands. Therefore, we invite you to submit a revised version of the manuscript that addresses the points raised during the review process. Please submit your revised manuscript within 30 days Jul 22 2025 11:59PM. If you will need more time than this to complete your revisions, please reply to this message or contact the journal office at plosntds@plos.org.  Please include the following items when submitting your revised manuscript:* A rebuttal letter that responds to each point raised by the editor and reviewer(s). You should upload this letter as a separate file labeled 'Response to Reviewers '. This file does not need to include responses to any formatting updates and technical items listed in the 'Journal Requirements' section below.* A marked-up copy of your manuscript that highlights changes made to the original version. You should upload this as a separate file labeled 'Revised Manuscript with Track Changes '.* An unmarked version of your revised paper without tracked changes. You should upload this as a separate file labeled 'Manuscript '. If you would like to make changes to your financial disclosure, competing interests statement, or data availability statement, please make these updates within the submission form at the time of resubmission. Guidelines for resubmitting your figure files are available below the reviewer comments at the end of this letter. We look forward to receiving your revised manuscript. Kind regards, Ben PascoeAcademic EditorPLOS Neglected Tropical Diseases Georgios PappasSection EditorPLOS Neglected Tropical Diseases

Shaden Kamhawi

co-Editor-in-Chief

Paul Brindley

co-Editor-in-Chief

 **Additional Editor Comments:**  Based on the revised manuscript and reviewer evaluations, I am pleased to inform you that your manuscript is close to being suitable for publication in PLOS Global Public Health. However, a few minor revisions are still required before we can proceed with final acceptance. These relate to map quality, terminology consistency, clarification of certain limitations, and minor grammatical and editorial fixes.

• Improve map resolution (Figure 1) and ensure all place names are legible.

• Fix remaining language/grammar issues, especially:

Line 361: “aware respondents” not “award”

Line 104: harmonise with abstract and clarify inclusion of environmental sampling.

• Clarify limitations further, ideally with a sentence explicitly noting the inability to assess AMR by farm type due to small subgroup sizes.

• Ensure uniformity of terminology for sample types throughout manuscript and figures (e.g., “poultry house floor sample” vs. “boot sock sample”).

• Final proofreading for style, tense consistency, and reference formatting.  **Journal Requirements:**  

1) We notice that your supplementary Tables are included in the manuscript file. Please remove them and upload them with the file type 'Supporting Information'. Please ensure that each Supporting Information file has a legend listed in the manuscript after the references list.

2) Please amend your detailed Financial Disclosure statement. This is published with the article. It must therefore be completed in full sentences and contain the exact wording you wish to be published.

1) State the initials, alongside each funding source, of each author to receive each grant. For example: "This work was supported by the National Institutes of Health (####### to AM; ###### to CJ) and the National Science Foundation (###### to AM).".

**Reviewers' comments:**  **Figure resubmission:**  While revising your submission, please upload your figure files to the Preflight Analysis and Conversion Engine (PACE) digital diagnostic tool, https://pacev2.apexcovantage.com/. PACE helps ensure that figures meet PLOS requirements. To use PACE, you must first register as a user. Registration is free. Then, login and navigate to the UPLOAD tab, where you will find detailed instructions on how to use the tool. If you encounter any issues or have any questions when using PACE, please email PLOS at figures@plos.org. Please note that Supporting Information files do not need this step. If there are other versions of figure files still present in your submission file inventory at resubmission, please replace them with the PACE-processed versions. **Reproducibility:**  To enhance the reproducibility of your results, we recommend that authors of applicable studies deposit laboratory protocols in protocols.io, where a protocol can be assigned its own identifier (DOI) such that it can be cited independently in the future. Additionally, PLOS ONE offers an option to publish peer-reviewed clinical study protocols. Read more information on sharing protocols at https://plos.org/protocols?utm_medium=editorial-email&utm_source=authorletters&utm_campaign=protocols

---

## [Editor Report · Decision Letter 2]

22 Jul 2025

Dear Dr. Kflewahd,

We are pleased to inform you that your manuscript 'Occurrence, Risk factors and Antimicrobial resistance of Campylobacter from Poultry and Humans in Central Ethiopia: A One Health Approach' has been provisionally accepted for publication in PLOS Neglected Tropical Diseases.

Best regards,

Ben Pascoe

Academic Editor

Georgios Pappas

Section Editor

Shaden Kamhawi

co-Editor-in-Chief

Paul Brindley

co-Editor-in-Chief

We appreciate the careful attention you have given to the reviewer and editorial comments. Your revised manuscript has addressed the scientific and presentation concerns, and we are pleased to inform you that it is now suitable for publication pending minor final adjustments:

Please update your Financial Disclosure statement to follow PLOS policy wording (remove placeholders such as “Grant N/A to Author Initials N/A”).

Ensure your Supporting Information files and legends are properly formatted according to PLOS guidance (legends placed after the references).

---

## [Editor Report · Acceptance letter]

Dear Dr. Kflewahd,

We are delighted to inform you that your manuscript, "Occurrence, Risk factors and Antimicrobial resistance of Campylobacter from Poultry and Humans in Central Ethiopia: A One Health Approach," has been formally accepted for publication in PLOS Neglected Tropical Diseases.

Best regards,

Shaden Kamhawi

co-Editor-in-Chief

Paul Brindley

co-Editor-in-Chief
